# Diversity Survey of a Pine Leafhopper Genus *Pinopona* Viraktamath & Sohi (Hemiptera: Cicadellidae: Koebeliini: Grypotina) in Yunnan Province, with Description of Two New Species

**DOI:** 10.3390/insects15120913

**Published:** 2024-11-22

**Authors:** Lin Lu, Yalin Zhang

**Affiliations:** Key Laboratory of Plant Protection Resources and Pest Management, Ministry of Education, College of Plant Protection, Northwest A&F University, Yangling 712100, China; lulin@nwsuaf.edu.cn

**Keywords:** Auchenorrhyncha, Deltocephalinae, Koebellinae, Grypotini, China

## Abstract

*Pinopona*, a genus of pine leafhoppers, belongs to the small tribe Koebelliini (Cicadellidae: Deltocephalinae) and can be distinguished by its crown featuring transverse striations along the fore margin; ocelli positioned below the anterior margin of the crown and distant from the eyes, rendering them not visible from above; a narrow anteclypeus that extends beyond the anterior margin of gena; and metatarsomere I having platellae on the plantar surface. Two new species, *Pinopona gongshanensis* and *Pinopona daliensis* spp. nov., are described and illustrated from Yunnan with a checklist and key for all species.

## 1. Introduction

The pine leafhopper genus *Pinopona* Viraktamath & Sohi, 1998 [1] was erected by the type species *Pinopona minuta* from India and Nepal and placed within the tribe Grypotini (Hemiptera: Cicadellidae Deltocephalinae). It was treated as a subtribe Grypotina of the tribe Koebeliini of Deltocephalinae by Dietrich and Dmitriev (2003) [2] in a recognized classification system based on the characteristics of a crown with few transverse fine striations on the anterior margin; short antennae, less than 1.5× the head length but not reaching half or more of the body length; ocelli below the anterior margin of the crown, not visible from above and distant from eyes; narrow and tapered anteclypeus beyond the anterior margin of the gena; and metatarsomere I platellae on the plant surface [2,3]. Both the genera *Shivapona* and *Sohipona* were transferred from Paraboloponini within Grypotini due to their characteristics of narrow and tapered anteclypeus and platellae on the plantar surface [4]. So far, this small subtribe Grypotina contains five genera and 10 species, including *Grypotellus* Emeljanov [5], *Grypotes* Fieber [6], *Pinopona* Viraktamath & Sohi [1], *Shivapona* Ghauri & Viraktamath [7], and *Sohipona* Ghauri & Viraktamath [7]. Investigating leafhoppers in Yunnan Province, the rare genus *Pinopona* was found in the southwest mountains of China and a phototaxis insect collected by light trap. In *Pinopona*, there are two known species: *P. sinae* (Stål, 1859) from China [8] and *P. minuta* Viraktamath & Sohi, 1998 from Pakistan, India, and Nepal [1]. The species diversity of *Pinopona* is not rich and mainly occurs the Himalayan region [2]. In this study, two new species, *Pinopona gongshanensis* sp. nov. from Gongshan and *Pinopona daliensis* sp. nov. from Dali, all from Yunnan Province, and the key to all species of this genus are provided.

## 2. Materials and Methods

Insect specimens were collected from Dali (100.01375 E, 25.74604 N) and Gongshan Mountains in Yunnan Province by net sweeping and light trap methods. Morphological terminology follows Dietrich (2005) [9], Dietrich and Dmitriev (2003) [2], and Zahniser and Dietrich (2013) [3]. Specimens were dissected and observed by stereo microscope (Leica Zoom 2000, Leica, Wetzlar and Germany), and pictures were taken with an advanced stereo microscope (Discovery V20, Zeiss, Macquarie Park, NSW, Australia) and CCD (AxioCam ICc5, Carl Zeiss, Göttingen and Germany) at the Insect Systematics and Biodiversity Platform, College of Plant Protection (NWAFU). The male genitalia of leafhopper species were utilized for identification and classification at the species level through dissection of the abdomen following soaking and boiling in a solution of 5–10% NaOH. Binomial classification method is adopted in taxonomic key. Type specimens of two newly identified species were deposited in the Entomology Museum of Northwest A&F University, China (NWAFU).

## 3. Results

### 3.1. Taxonomy

#### 3.1.1. Tribe Character

Tribe Koebeliini Baker, 1897= Grypotini Haupt, 1929 [10]Type genus: *Koebelia* Baker, 1897 [11].Note: the character of tribe can be refered to Zahnier & Dietrich’ 2013 [3].

#### 3.1.2. Subtribe Character

Subtribe Grypotina Haupt, 1929 [10]Type genus: *Grypotes* Fieber, 1866 [6].

Diagnosis. (Modified from Zahniser and Dietrich, 2013 [3].) Body pale greenish to yellow, or dark brown, small to medium size. Crown broader than pronotum, rounded to face with transverse striations or carinate; clypeus long, narrow and extending well beyond the genera; ocelli distant from eyes, located below anterior margin of crown, not visible from above. Forewing with appendix restricted to anal margin; forewing veins not pustulate; metafemur apex macrosetae 2 + 2 + 1; metatarsomere I with platellae on the plantar surface.

Distribution. Palearctic, Oriental, and Nearctic regions.

#### 3.1.3. Generic Character

*Genus Pinopona* Viraktamath & Sohi, 1998*Pinopona* Viraktamath & Sohi, 1998: 114 [1].Type species: *Pinopona minuta* Viraktamath & Sohi, 1998 [1].

Diagnosis. Body yellowish or brown. Crown with faint rufous or black serpentine patches on the front. Pronotum with irregular russet or dark brown markings. Scutellum with basal angle reddish-brown. Forewing greenish, with apex smoky and hyaline, veins yellow.

Head broader than pronotum; anterior margin thick in profile, fore margin with few fine transverse striations. Crown sub-parallel, round with coronal suture present, mid-length slightly shorter than the half width between eyes, slightly depressed in discal area, with fine longitudinal striations and fore margin slightly elevated. Ocelli large, below fore margin, not visible from above, distant from corresponding eyes. Face with lateral frontal sutures divergent dorsad of antennal pits, extending to ocelli; clypeal suture straight; anteclypeus narrow and tapered with lateral margins subsinuate, beyond the margin of gena, and apex slightly subacute and inward; lora slightly narrower than basal width of anteclypeus; gena slightly emarginate to straight below eyes; antennae 1.5× less than head length; antennal pits situated near upper corner of eyes, encroaching onto clypeus; antennal ledges weakly carinate. Pronotum slightly produced between eyes, hind margin slightly concave and equal to scutellum in length; lateral margin short with carinae vestigial; scutellar suture arcuate. Forewing with four apical cells and three subapical cells, inner subapical cell open; appendix relatively narrow; with discrete punctations at the base of Sc and R veins and vestigial at the base of claval region; without cross vein between A1 and A2, also absent between A1 and claval suture. Fore femur with AM1 stout and long, AV1 fine and long, with several stout AV setae, IC = 2 or 5. Fore tibia with dorsal setal arrangement AD and PD 4 + 4. Metafemur macrosetae 2 + 1 + 1. Metatarsomere I not expanded with five platellae.

Male genitalia. Pygofer lobe produced narrowing to apex with numerous macrosetae subapically, with dorsal bridge sclerotized and dorsal appendage extended obliquely ventrad and ventral margin relatively straight with ventral appendage. Valve broad, triangular. Subgenital plate broad, triangular with two rows of macrosetae along lateral margin and near middle subdistal. Style apophysis extended, strongly laterally with apex strongly sculptured, and preapical apophysis sub-rectangular. Connective Y-shaped, with stem slightly shorter than arms and articulate with the preatrium of aedeagus. Aedeagus dorsal apodeme developed, and with shaft compressed and curved dorsad, smooth or dark scaly punctations at apex laterally, with pairs of apical reflexed processes curved ventrad, sheep-horn-like, extending from a quarter to a half distance of shaft, gonopore apical on ventral surface. Segment X somewhat sclerotized laterally, longer than wide.

Distribution. China (Hunan, Hongkong, Yunnan); Pakistan, India and Nepal.

Host plant. *Pinus* spp. (Pinaceae).

Remarks. This genus is more similar to other genera within the subtribe Grypotina than Paraboloponina due to its characteristics of ocelli distant from eyes, not visible from above; its antennae comparatively short, 1.5× less than head length, not reaching half body length; narrow and tapered anteclypeus and platellae on plantar surface. The genus comprises four species: *P. minuta* Viraktamath & Sohi [1], *P. sinae* (Stål) [8], *P. gongshanensis* sp. nov., and *P. daliensis* sp. nov. (in checklist). This genus can be differentiated from other genera by its metafemur seta formulation 2 + 1 + 1 and slender aedeagus. Two new species differ from two already known species, namely *P. sinae* and *P. minuta*, in having an aedeagal shaft much more slender and comparatively long, differently curved apical process of the aedeagal shaft which also differ in their relative lengths, more distally narrowed and longer apophysis of the style.

#### 3.1.4. Checklist All Species of *Pinopona*

*P. minuta* Viraktamath & Sohi, 1998: 114. Pakistan, India and Nepal [1].*P. sinae* (Stål, 1859: 293) [8]; Dietrich & Dmitriev, 2003: 773 [2]. China (Hunan, Hongkong).*P. gongshanensis* sp. nov. China (Yunnan).*P. daliensis* sp. nov. China (Yunnan).

#### 3.1.5. Key to Male Species of *Pinopona*

With apical processes of aedeagus convergent(Figure 37, Dietrich & Dmitriev, 2003 [2])………………………………..*P. sinae*With apical processes of aedeagus divergent (Figure 1P and Figure 2Q)……2With some dark scaly punctations from apex to a quarter of shafton lateral surface (Figure 1M,Q)………………………*P. gongshanensis* sp. nov.Without dark scaly punctations on lateral shaft (Figure 2N)………………….3More distally narrowed and longer apophysis of the style; aedeaguswith apical processes and shaft comparatively long (Figure 2M,P,Q)…………………………………………………………………...*P. daliensis* sp. nov.Relatively shorter apophysis of the style; aedeagus with apicalprocesses and shaft comparatively short(Figures 11 and 12, Viraktamath & Sohi, 1998 [1]) ……..*P. minuta*

### 3.2. Species Description

#### 3.2.1. *Pinopona sinae* (Stål, 1859)

*Jassus (Thamnotettix) sinae* Stål, 1859: 293 [8].

*Thamnotettix sinae* Metcalf, 1967: 776 [12].

*Pinopona sinae*, Dietrich & Dmitriev, 2003: 773 [2].

Diagnosis. Body yellow with numerous brown irregular longitudinal striations. Small size, crown broader than pronotum, with anterior margin weakly produced rounded, disc flat, a few irregular longitudinal striations. Ocelli located under anterior margin and distant from eyes, the distance nearly equal to the distance from coronal suture, not visible from above; anteclypeus tapered to apex. Forewing with apex developed. Male pygofer tapered to apex with macrosetae on lateral lobe. Valve broad triangular. Style apophysis digitate, apex slightly truncate. Connective Y-shaped, lateral arms equal to the length of stem. Aedeagal shaft smooth, curved dorsad, with pair of apical reflexed processes curved ventrad, gonopore apical on ventral surface.

Material examined. 15♂♂12♀♀, Hunan Prov., Hengshan mountains, 1985.VIII.11, Zhang YL & Chai YH (NWAFU).

Distribution. China (Hunan, Hongkong).

Remarks. This species can be distinguished from *P. gongshanensis* and *P. daliensis* by its relatively short aedeagal shaft and convergent apical processes of the aedeagus.

#### 3.2.2. *Pinopona gongshanensis* sp. nov.

(urn:lsid:zoobank.org:act:ED749C8E-1A50-4127-BD50-59605705951F)

Figure 1A–S

Description. Length: ♂4.5–4.8 mm. Body yellowish. Crown with faint rufous serpentine patches on the front. Pronotum with irregular russet markings. Face yellowish with an arched rufous band between antennal pits. Forewing yellowish, with apex smoky and hyaline, veins greenish. Forefemur with IC = 5.

Male genitalia. Pygofer with ventral weak appendage directed posteriorly. Aedeagus with paired apical reflexed processes extended ventrad to a quarter of shaft, relatively short, with some dark scaly punctations from apex to a quarter of shaft on lateral surface.

Material examined. Holotype:♂, China, Yunnan Province, Gongshan County, Bingzhongluo countryside, 1700 m, 12.vi.2019, light trap, coll. Lin Lu (NWUAF). Paratypes: 4♂, same data as holotype (NWUAF).

Etymology. The species is named after its locality: Gongshan County.

Remarks. This species is similar to *P. daliensis* in shape, but can be differentiated from the latter by the yellow coloration in habitus and the male genitalia characteristic of aedeagus with relatively short apical processes extending to a quarter of shaft.

#### 3.2.3. *Pinopona daliensis* sp. nov.

(urn:lsid:zoobank.org:act:B2F4B7D8-37E7-4782-939B-01687F51CD97)

Figure 2A–Q

Description. Length: ♂4.8–5.0 mm. Body brownish. Crown with dark brownish serpentine patches on the front. Pronotum with irregular black markings on anterior margin. Scutellum with black angular markings. Face brownish with arched black band between antennal pits and few black spots on lora and anteclypeus. Forewing greenish, with apex smoky, veins greenish. Forefemur with IC = 2.

Male genitalia. Pygofer with ventral weak appendage directed posteriorly. Aedeagus with paired apical reflexed processes slightly divergent, small pointed round at apex in caudal view and extended ventrad to half of shaft, relatively long, without dark scaly punctations from apex to half of shaft in lateral surface.

Material examined. Holotype:♂, China, Yunnan Prov., Dali City, Yangbi County, Pingpo Township, Shangpingpo countryside, 100.01375 E, 25.74604 N, 2664 m, 18.vii.2019, light trap, coll. Hu Li (NWUAF). Paratypes: 1♂, same data as holotype (NWUAF).

Etymology. This species is named from its locality: Dali City.

Remarks. This species differs from *P. gongshanensis* by brownish coloration in habitus and the character of the strong ventral appendage of pygofer lobe and aedeagus with relatively long apical processes extending to half of shaft.

## 4. Discussion

Species diversity of leafhopper in Gaoligong Mountain and its surrounding areas (Diancang Mountain) was investigated with description and illustration of two new species. One of them is *Pinopona gongshanensis* from Gongshan county in the southeastern margin of the Qinghai–Tibet Plateau and the hinterland of the Gaoligong Mountains, and the other species *P. daliensis* is from Dali township, located at the intersection of the Yunnan–Guizhou Plateau and the Hengduan Mountains; habitats of both species are at high altitudes of more than 1700 m. As a result, the species diversity of this genus is not high and is mainly confined to the Himalayan region. Future research will focus on the geographical distribution pattern of this genus, which supports the adaptative evolution evidence of the Tibetan Plateau uplift and unstable Ice Age.

## 5. Conclusions

All four species of the rare pine leafhopper genus *Pinopona* Viraktamath & Sohi are reported, with descriptions and illustrations of two new species from China (Yunnan), i.e., *P. gongshanensis* and *P. daliensis*. In addition, the article determines the Himalayan fauna of the tribe Koebeliini and enriches the leafhopper species diversity of China.

## Figures and Tables

**Figure 1 insects-15-00913-f001:**
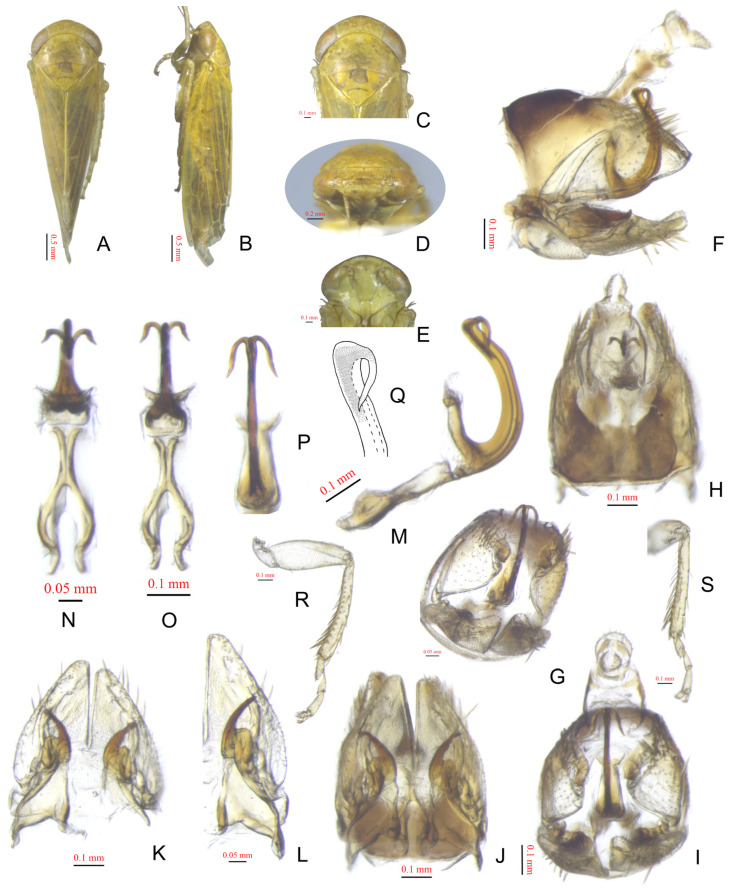
*Pinopona gongshanensis*. (**A**,**B**) Male, dorsal and lateral view; (**C**,**D**) head, dorsal and anterodorsal view; (**E**) face; (**F**–**J**) pygofer, lateral, caudolateral, dorsal, caudal, and ventral view; (**K**) valve, subgenital plates and styli, ventral view; (**L**) style and subgenital plate, dorsal view; amplification and aedeagus, (**M**) lateral view, (**N**) dorsal view, (**O**) ventral view, (**P**) caudal view; (**Q**) apex of aedeagus, lateral view; (**R**,**S**) forefemur and foretibia, anterior and dorsal view.

**Figure 2 insects-15-00913-f002:**
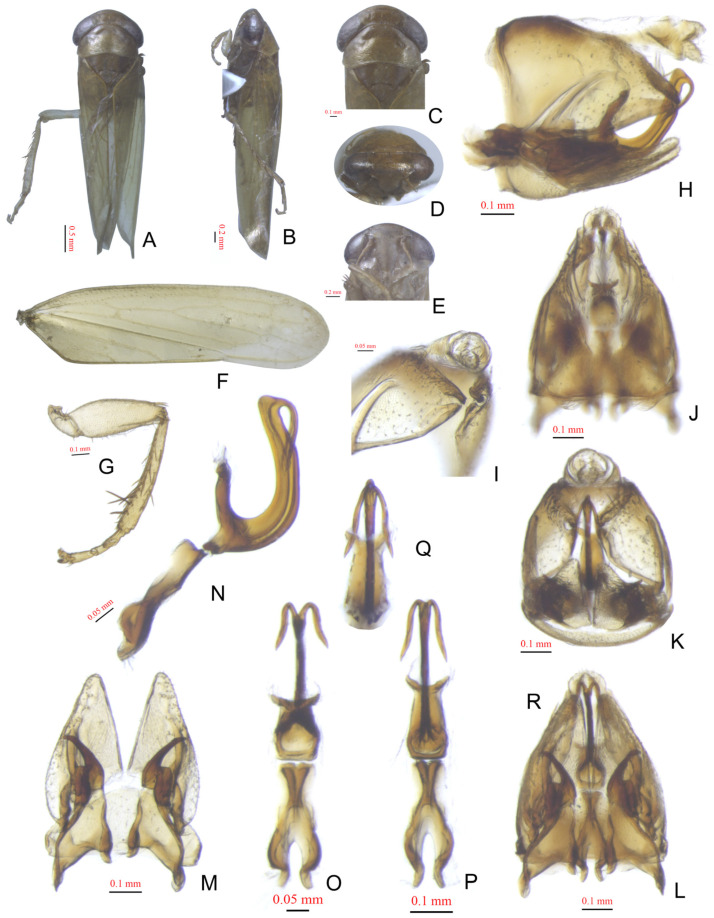
*Pinopona daliensis*. (**A**,**B**) Male, dorsal and lateral view; (**C**,**D**) head, dorsal and anterodorsal view; (**E**) face; (**F**) forewing; (**G**) forefemur and foretibia, anterior view; (**H**–**L**) pygofer, lateral, caudolateral, dorsal, caudal, and ventral view; (**M**) valve, subgenital plates and styli, dorsal view; aedeagus and amplification, (**N**) lateral view, (**O**) dorsal view, (**P**) ventral view, (**Q**) caudal view.

## Data Availability

No new data were created or analyzed in this study. Data sharing is not applicable to this article.

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
