# Peer review of "Diversity Survey of a Pine Leafhopper Genus Pinopona Viraktamath & Sohi (Hemiptera: Cicadellidae: Koebeliini: Grypotina) in Yunnan Province, with Description of Two New Species"

_insects, 2024, doi:10.3390/insects15120913_

Round 1
Reviewer 1 Report
Comments and Suggestions for Authors
Lines 53-58 : What methodology was used to prepare the terminalia for dissection? Cite the methodology. What method was used to collect insects? What is the collection location? Add the name of the collection city and the geographic coordinates of the collection point. Add to the methodology that was produced a new key to the genus.
Lines 60-127 : Add the number of individuals studied and information on the deposit location of the two species.
Lines 150-203 : Add this excerpt from the work describing the new species above line 129 (checklist) and below line 128.
Reviewer 2 Report
Comments and Suggestions for Authors
Page 3, line 117. Distribution of the genus is China (Yunnan), which is incorrect.
Descriptions of new species are very minimalistic. Not even body size of the insects is provided. I would recommend to provide a better descriptions.
Diagnosis. Two newly described species are compared to each other. When new species are described it is advised to compare them with previously known species.
I would strongly recommend to add line drawings for new species to supplement photographs.
Round 2
Reviewer 2 Report
Comments and Suggestions for Authors
NA